# Respiratory Syncytial Vaccination: Parents’ Willingness to Vaccinate Their Children

**DOI:** 10.3390/vaccines12040418

**Published:** 2024-04-15

**Authors:** Vincenza Sansone, Silvia Angelillo, Francesca Licata, Grazia Miraglia del Giudice, Gabriella Di Giuseppe

**Affiliations:** 1Department of Experimental Medicine, University of Campania “Luigi Vanvitelli”, 80138 Naples, Italy; vincenza.sansone@unicampania.it (V.S.); grazia.miragliadelgiudice@unicampania.it (G.M.d.G.); 2Department of Health Sciences, University of Catanzaro “Magna Græcia”, 88100 Catanzaro, Italy; silvia.angelillo@studenti.unicz.it (S.A.); f.licata@unicz.it (F.L.)

**Keywords:** children, cross-sectional study, Italy, parents, RSV, vaccine, willingness

## Abstract

Background: This study was conducted to assess parents’ willingness to vaccinate their children with the RSV vaccine and the key predictors of this intention among parents in Italy. Methods: Data were collected using an anonymous self-administered questionnaire from April to November 2023, targeting parents in public kindergartens and nursery schools in southern Italy. The survey assessed parents’ socio-demographic characteristics, health-related details, their child’s health status, attitudes toward RSV infection and its vaccine, and their source(s) of information. Results: A total of 404 parents agreed to participate in the study. Only 18.2% of participants were very concerned that their children could get infected by RSV, and this concern was more likely among parents whose child had been diagnosed with bronchiolitis, those who received information from HCWs, those who had heard of RSV, and those who needed additional information. Almost half (51.3%) were willing to vaccinate their child, and this inclination was more likely among fathers, employed parents, those with daughters, those who had heard of RSV, those who received information from HCWs, and those who needed additional information. Conclusions: An educational campaign regarding a future RSV vaccine, especially about its safety and efficacy, is needed in order to improve parents’ willingness.

## 1. Introduction

Acute lower respiratory infections continue to be one of the leading causes of death worldwide among children under five years of age, accounting for approximately 13.1% of annual deaths [1]. The most common viral cause of these infections is respiratory syncytial virus (RSV) [2], which contributes considerably to worldwide morbidity and mortality. In particular, in infants and young children, the first infection may cause severe bronchiolitis, which can sometimes be fatal [3]. One out of every fifty deaths in children aged 0–60 months and one out of every twenty-eight deaths in children aged 28 days to 6 months is attributed to RSV [4]. In older children without comorbidity, repeated infections of the upper respiratory tract are common and range from subclinical infection to symptomatic upper respiratory tract disease [3]. This virus is transmitted by direct and indirect contact, is highly contagious, and causes frequent reinfections in children [5,6,7].

Moreover, it has been established that children who were hospitalized for RSV infections within the initial two years of life have an increased likelihood of asthma-related hospitalization [8]. Additionally, the clinical manifestation of RSV infection can vary from a mild to a very serious form of infection, requiring hospitalization [9,10]. Furthermore, children have a higher risk of acquiring RSV in comparison to other respiratory viruses, such as influenza [11], and this emphasizes the substantial impact of RSV on healthcare services and highlights the need to implement effective preventive measures to mitigate its burden.

Since there is no effective and targeted treatment for RSV infections, it is essential to implement preventive measures to reduce associated morbidity and mortality. In Italy, monoclonal antibody therapy is recommended only for children at higher risk of infection. Several ongoing clinical trials are evaluating the safety and immunogenicity of an RSV vaccine among children up to five years of age [12,13]. Moreover, the European Medicines Agency (EMA) has already recommended the authorization of a vaccine among pregnant women to protect their infants up to six months of age through passive immunization [14]. 

Vaccinating children up to five years of age could have a substantial effect on reducing RSV infection’s burden. Since an RSV vaccine targeting children is set to be included as part of the childhood immunization program, it is important to establish the willingness of parents to vaccinate their children. Specifically, it is necessary to assess the barriers (age, low level of education, economic status, concern for the safety of vaccines) and the facilitators (appropriate sources of information and trust in healthcare professionals) that may affect the parents’ willingness to vaccinate their children. In addition, these results could be used to better target the development of approaches aimed at increasing the acceptance of the RSV vaccine. These factors have been studied for other recommended vaccinations [15,16,17,18], but few surveys have been conducted to explore the acceptance of a future RSV vaccine among pregnant women [19,20] and healthcare workers (HCWs) [21]. Therefore, the primary outcome of this study was to assess parents’ willingness to vaccinate their children with the RSV vaccine and to identify the key predictors of this intention.

## 2. Materials and Methods

### 2.1. Setting and Study Population

This survey is part of a larger project that investigated perceptions and acceptance towards the RSV vaccine among different population groups in southern Italy, including pregnant women [19]. Data were collected from April to November 2023 in randomly selected public kindergartens and nursery schools in the geographic areas of Naples and Catanzaro, southern Italy. The survey included parents/guardians of children up to five years of age. If parents/guardians had multiple eligible children, they were asked to provide answers regarding the youngest child.

The minimum sample size was calculated assuming that 60% of parents would be willing to vaccinate their children against RSV (primary outcome) [19]; considering a margin of error of 5% and a confidence interval of 95%, the “minimum” total sample size was estimated to be 369 parents.

### 2.2. Data Collection

This survey was approved by the Ethics Committee of the Teaching Hospital of the University of Campania “Luigi Vanvitelli” (protocol number 0016978/i). Prior to beginning the investigation, the research team sought approval for participation from the heads of the chosen kindergartens and nursery schools. Once approval was received, the research team, in collaboration with teachers, delivered an envelope to each eligible parent/guardian when they picked up their child from school. The envelope contained an invitation letter with the survey’s purpose and procedures, an informed consent form, the questionnaire, and two additional envelopes for returning the completed questionnaire and the signed consent form, separately. The letter ensured confidentiality and anonymity for all collected data, underlined that participation was voluntary, and informed participants of their right to withdraw at any stage without providing a reason. To enhance response rates, the research team applied a reminder strategy involving multiple follow-ups, actively engaging teachers to encourage participation, and gathering data from non-respondents. No monetary incentives were offered for participation. 

### 2.3. Questionnaire

The self-administered questionnaire was prepared by the research team, adapted and modified from a previously published study of the research group on this topic [19]. It was structured in three sections for a total of 24 questions. The first section aimed to gather socio-demographic information and health-related details of the parents, covering data on age, gender, marital status, education level, employment status, number of children, and underlying chronic medical conditions and on the selected child, including age, gender, chronic medical conditions, eventual preterm birth, or diagnosis of bronchiolitis.

The second section consisted of three questions assessing participants’ awareness of RSV infection, the sources from which they obtained information that could be selected from a list of 7 choices, and their perceived need for additional information. Before proceeding with the survey, respondents who were unaware of RSV infection were provided a brief explanation. 

The third section comprised three questions about attitudes toward RSV infection and vaccination. Two questions, measuring parents’ concern that their child might get infected by RSV and the perceived usefulness of the RSV vaccine administration for their children, were scored on a 10-point Likert scale, with higher grades corresponding to a better attitude. The remaining question investigated the willingness to vaccinate their child against RSV, with “yes”, “no”, and “do not know” as response options. Reasons for their intention were explored through a closed-ended multiple-choice question from a list of 6 options, allowing multiple responses. The questionnaire required approximately 10 min to be completed.

A pilot study was carried out on a random sample of fifty parents to ensure the correct interpretation, feasibility, and reliability of the questions. Participants were instructed to complete questionnaires and to report written feedback. Since there were no significant issues related to the survey used, no changes were made to the questionnaire and the results obtained in the pilot study were integrated into the analysis.

### 2.4. Statistical Analysis

Primarily, descriptive statistics were performed to determine the distributions of both qualitative and quantitative variables. Qualitative variables were presented as numbers and proportions, while quantitative variables were expressed as means and standard deviations. A multivariate linear regression model exploring the variables independently associated with parents’ concern that their child could get infected by RSV (secondary outcome) (Model 1) (10 point scale from 1 “not at all concerned” to 10 “very much concerned”) and a multivariate logistic regression model estimating the variables independently associated with willingness to vaccinate their child against RSV (primary outcome) (Model 2) (0 = no/uncertain; 1 = yes) were developed. Independent variables involving several categories were dichotomized as subsequently described. The following independent variables related to the responding parent were examined for both models: age, in years (continuous), gender (male = 0; female = 1), marital status (unmarried/separated/divorced/widowed = 0; married/cohabiting with a partner = 1), both parents with a university degree (no = 0; yes = 1), both parents employed (no = 0; yes = 1), having a chronic medical condition (no = 0; yes = 1), having more than one child (no = 0; yes = 1), having heard of RSV infection (no = 0; yes = 1), having received information about RSV from healthcare workers (HCWs) (no = 0; yes = 1), need for additional information (no = 0; yes = 1). The independent variables concerning the selected child were the following: age, in years (continuous), gender (male = 0; female = 1), firstborn (yes = 0; no = 1), having a chronic medical condition (no = 0; yes = 1), preterm born (no = 0; yes = 1), having been diagnosed with bronchiolitis (no = 0; yes = 1), and having been hospitalized because of a respiratory infection (no = 0; yes = 1).

The model building strategy involved two steps. For the linear regression model (Model 1) the first step involved the performance of univariate analysis for each of the independent variables that could influence parents’ concern that their child could get infected by RSV. Specifically, a *t*-test was performed for binary qualitative variables and Pearsons’ correlation coefficient was calculated for quantitative variables. For the logistic regression model (Model 2), the first step involved the performance of several univariate logistic regressions for each of the independent variables that could potentially influence the willingness to vaccinate their child against RSV, and crude odds ratios (ORs) and related confidence intervals (CIs) were calculated. For both models, the variables in the univariate analysis that showed a *p*-value ≤ 0.25 were included in the respective multivariate model. Adjusted ORs with a 95% CI and standardized regression coefficients (*β*) for potential determinants were estimated in logistic and linear regression models, respectively. Statistical significance was determined by a two-sided *p*-value of 0.05 or less. The statistical analysis was conducted using STATA software version 18.

## 3. Results

### 3.1. Socio-Demographic, General, and Health-Related Characteristics

Of the 667 invited parents, 404 decided to participate in the study with a response rate of 60.6%, and the main respondents’ characteristics are shown in Table 1. Most (81.5%) were mothers, the average age was 34.5 years, most were married or cohabiting (63.9%), almost half had a university degree (45.2%), 70.6% were employed, and 52% had only one child. Two-thirds of respondents had an employed partner (89.9%) and 41.6% had a partner with a university degree. Chronic medical conditions were reported in 12.6% of the respondents and 7.5% of cohabitants. In respect to children’s characteristics, 59.3% were firstborn, 52.6% were males, 12.8% were preterm born, and only 1.8% had a chronic medical condition. Regarding respiratory illness, 12.2% of the respondents reported that their children had been diagnosed with bronchiolitis and 8% were hospitalized because of a respiratory infection.

### 3.2. Parents’ Attitudes Regarding RSV Infection and Vaccination

When parents responded about their attitudes, 18.2% of them expressed a high level of concern that their children could get infected by RSV, with an overall mean value of 6.4 out of a maximum score of 10. As described in Table 1, the association of several variables with the outcomes of interest was found in the univariate analyses, and it has been partially confirmed by multivariate analyses (Table 2). The stepwise linear regression analysis, performed to estimate predictors of the concern that their child could get infected by RSV, revealed that parents whose child had been diagnosed with bronchiolitis, those who received information from HCWs, who had heard of RSV, and who needed additional information were significantly more concerned (Model 1 in Table 2).

Furthermore, RSV vaccine administration for children was believed very useful to protect them by only 18.2% of the sample, with an overall mean value of 6.1 out of a maximum score of 10. 

As regards the acceptance of an RSV vaccine, 51.3% of parents expressed their willingness to vaccinate their child. The stepwise logistic regression analysis showed that fathers, those who had a daughter, those who had heard of RSV, those had received information from HCWs, and those who needed additional information were more likely to be willing to vaccinate their child (Model 2 in Table 2). Among the most frequent reported reasons for the acceptance of the vaccine there were the need to protect their children (66.5%), being concerned about vaccine safety (41%), having been recommended by a pediatrician (38.5%), trust in vaccines (36%), believing in the effectiveness of the RSV vaccine (30%), and concerns regarding the severity of RSV infection (17.5%). Among the reasons for refusal, concern about possible side effects (59.6%) was the most reported, followed by vaccine safety (25.3%), a lack of trust in vaccinations (22.6%), doubts about the effectiveness (17.8%) and utility of the vaccine to protect the child (15.7%), and belief that RSV infection is not a cause of severe disease (1.4%).

### 3.3. Sources of Information

More than half (57.6%) of parents had heard about RSV infection, and 44.8% of them had acquired information from HCWs, 34.9% from family members or friends, and 23.4% from TV/journals. Moreover, almost two-thirds of the respondents (64.7%) reported the need to receive additional information about a vaccine against RSV for their child.

## 4. Discussion

The current survey, which is part of a large project regarding the willingness to vaccinate against RSV among different groups of individuals [19], focused on knowledge and attitudes of parents of children up to five years of age and adds motivating and useful information on this topic.

First, our sample, with 12.2% of children having had bronchiolitis, confirmed that this respiratory infection is common in Italy and this is in line with the last Italian national report of April 2024, describing that in the previous months RSV infection was the second most common respiratory infection after influenza [22]. This trend is also described in a recent Italian study, where an increased number of total RSV cases was associated with the limitation measures adopted during the COVID-19 pandemic, which had restricted the transmission of respiratory viruses [23]. Moreover, an increased number of hospitalizations caused by RSV infection has been described worldwide, with more severe diseases in infants compared with previous periods, explained by a more virulent infection [24,25].

More than half of respondents (57.6%) reported having heard about RSV infection, and this proportion is larger compared to the population of pregnant women in Italy (32.2%), England (29%), and Australia (17%) [19,26]. Moreover, although information about RSV appears to be well-known worldwide compared to other respiratory infections, there is a lower level of knowledge among parents [27]. This suggests that there is a need to promote information campaigns on RSV and its modes of transmission. HCWs play a fundamental role and indeed they were indicated as the most frequent source of information on this topic (44.8%). Therefore, it is essential to support HCWs in their role of informing parents about how to reduce the transmission of respiratory infections. Indeed, it has been reported that many parents did not have information or good knowledge about bronchiolitis caused by RSV [28]. In this regard, the information transferred by hospital staff at birth or during the first visits of the newborn, for example, with a brochure, may have a significant value in reducing the spread of infection.

Second, only half (51.3%) were willing to vaccinate their child. This result is significantly lower than the willingness reported in the USA, where 71.1% of parents were planning to vaccinate their children against RSV [29]. It is interesting to underline that, in a similar population, the willingness reported regarding the COVID-19 vaccine was higher (65.5%), whereas that regarding influenza was similar (48.6%) compared with the present findings [30,31]. A possible explanation for the lower intention of parents to vaccinate their children against RSV can be linked with another important result explored regarding the attitudes. Indeed, only 18.2% of parents were very concerned that their children could be infected by RSV and considered the vaccine very useful to protect their children. Key motivators of intention to adhere to future RSV vaccination for their children included protecting them and having a recommendation from a pediatrician. Concerns about the side effects of the vaccine were shared by almost two-thirds of parents who did not intend to vaccinate or were uncertain about vaccinating their children. These results agree with the existing literature, in which the most frequently cited reason for accepting a new vaccination is to protect their children, whereas for refusal it is the perceived risk of adverse effects [32,33,34,35]. Moreover, receiving information from HCWs is associated with caregivers’ lower hesitancy to vaccinate children for other respiratory diseases, for example, influenza or COVID-19 [36,37]. In the case of a new vaccine against RSV, therefore, it is necessary to plan educational interventions aimed at HCWs, especially pediatricians, to enable them to learn the necessary information to be transferred to parents. Examples of educational strategies could be face-to-face and online courses to increase vaccine knowledge or offering interactive approaches with roleplay sections to support comfort level and perceived effectiveness when counseling vaccine-hesitant parents [38].

Third, the findings of the multivariate regression analysis indicated that receiving information from HCWs, having heard of RSV, and the need for additional information are independently identified as significant determinants of both outcomes of interest. One possible explanation is that concern and willingness are influenced by having more information about the disease. Furthermore, it appears that knowledge and the need for information are fundamental to increase the RSV vaccination willingness among the target population, as reported for another new vaccine against a respiratory disease and a future RSV vaccination for newborns in Italy [19,39]. Aligned with these results, having a child diagnosed with bronchiolitis was a predictor of parents’ concern. A possible key interpretation is that parents who have already had experience with bronchiolitis know the possible risks related to the disease and, therefore, are much more concerned for their child. Moreover, two socio-demographic characteristics, being a father and having a daughter, were found to be key predictors of parents’ willingness to vaccinate their child. In particular, fathers’ willingness is consistent with the gender difference seen among caregivers to vaccinate their children with a new vaccine during the COVID-19 pandemic [40,41,42]. The reasons explaining why fathers are more likely to vaccinate should be investigated in order to reduce hesitation among mothers, who are often the main caregivers of young children. Moreover, parents of daughters are more likely to vaccinate them. Gender differences in vaccination uptake in the adult population have been reported [43], and attention to gender-related issues in immunization programs, with the goal of reducing gender-related barriers to access to vaccination specifically addressed to female children, is considered a priority in the international literature, especially in low-income countries [44]. The results of this study run counter to the literature and are worth further investigation.

This study presents several limitations due to the study design. First, as this is a cross-sectional study, it is only possible to observe correlations and not the attribution of cause and effect due to the temporal relationship between the determinants and the different outcomes of interest. Second, regarding attitudes, respondents may have answered in the most “desirable” way, and this may have resulted in an overestimation of the positive attitudes toward the willingness to vaccinate their children against RSV. To solve this limitation, the questionnaire was self-administered and, in this way, the researchers have ensured the confidentiality and anonymity of responses and data collection. Third, since a history of respiratory infections was investigated, a recall bias may have occurred, overestimating or underestimating this information. Fourth, the majority of the sample was composed of mothers, and fathers’ opinions were not entirely known or evaluated. Moreover, the study was carried out only in two regions of southern Italy and this affected the generalizability of the findings. Finally, since the questionnaire was completed at home, parents could have retrieved information regarding the topic of the survey before responding to the questionnaire. Despite these limitations, a sufficient response rate was achieved, and the findings of this study provide important insights and new knowledge on a public health issue that has a relevant influence.

## 5. Conclusions

In conclusion, this survey shows a lack of knowledge about RSV infection among parents. To improve parents’ willingness and support HCWs in promoting this future vaccine, educational campaigns on the safety and efficacy of a future RSV vaccine are necessary. High vaccination coverage is fundamental to control the global RSV infection spread described in the child population. Thus, it is important to understand which are the key determinants affecting parents’ decision to vaccinate their children and help health policymakers. This information can be used for evidence-based targeted campaigns and health interventions to finally improve future RSV vaccine uptake among children. There is need to develop vaccine confidence through clear messages and effective community engagement. Targeted public health strategies should aim to moderate parents’ concerns regarding the RSV vaccine to improve their willingness. Future research is needed to investigate the willingness to vaccinate against RSV in other eligible populations, such as older people, especially when the vaccination will be available in Italy.

## Figures and Tables

**Table 1 vaccines-12-00418-t001:** Results of univariate analysis investigating the determinants associated with parents’ concern that their child could get infected by RSV and willingness to vaccinate their child against RSV in the study population.

Characteristics	Total (N = 404)		Parents’ Concern That Their Child Could Get Infected by RSV (10-Point Scale from 1 “Not at All Concerned” to 10 “Very Much Concerned”)	Willingness to Vaccinate Their Child against RSV (0 = No/Uncertain; 1 = Yes)	
**Parent**	**n**	**%**	**Mean ± DS**	**n**	**%**
Age, years	34.5 ± 4.9 ^b^				
			Pearson correlation coefficient (df = 382) = 0.07, *p* = 0.16	1.02 (0.98–1.07) ^c^	
Gender (399) ^a^					
Female	325	81.5	6.3 ± 2.8	150	75.4
Male *	74	18.5	6.2 ± 2.6	49	24.6
			*t*-test (383) = −0.39, *p* = 0.69	0.44 (0.026–0.76) ^c^	
Marital status (402) ^a^					
Unmarried/separated/ divorced/widowed *	26	6.5	6.2 ± 3.3	12	6
Married/cohabiting	376	93.5	6.3 ± 2.7	188	94
			*t*-test (385) = −0.01, *p* = 0.929	1.25 (0.56–2.77) ^c^	
Both parents with a university degree (377) ^a^					
No *	254	67.3	6.1 ± 2.9	111	57.2
Yes	123	32.6	6.9 ± 2.3	83	42.8
			*t*-test (362) = −2.67, *p* = 0.008	2.73 (1.72–4.33) ^c^	
Both parents employed (374) ^a^					
No *	124	33.2	5.8 ± 2.9	42	22.1
Yes	250	66.8	6.6 ± 2.6	148	77.9
			*t*-test (359) = −2.79, *p* = 0.006	2.89 (1.83–4.56) ^c^	
Chronic medical condition					
No *	353	87.4	6.2 ± 2.8	170	84.6
Yes	51	12.6	6.9 ± 2.4	31	15.4
			*t*-test (387) = −1.69, *p* = 0.091	1.56 (0.85–2.84) ^c^	
Number of children					
1 *	210	52	6 ± 2.7	121	60.2
>1	194	48	6.7 ± 2.8	80	39.8
			*t*-test (387) = −2.33, *p* = 0.021	0.56 (0.38–0.84) ^c^	
Ever heard about RSV (403) ^a^					
No *	171	42.4	4.9 ± 2.8	50	24.9
Yes	232	57.6	7.4 ± 2.3	151	75.1
			*t*-test (387) = −9.99, *p* < 0.001	5.06 (3.28–7.81) ^c^	
Having received information from HCWs (403) ^a^					
No *	299	74.2	5.7 ± 2.8	126	62.7
Yes	104	25.8	8 ± 2	75	37.3
			*t*-test (387) = −7.76, *p* < 0.001	3.93 (2.36–6.53) ^c^	
Need for additional information (367) ^a^					
No *	138	37.6	5.4 ± 3.1	31	15.9
Yes	229	62.4	6.8 ± 2.5	164	84.1
			*t*-test (365) = −4.89, *p* < 0.001	8.71 (5.32–14.25) ^c^	
**Selected child**	**n**	**%**	**Mean ± DS**	**n**	**%**
Age, years	2.9 ± 1.3 ^b^				
			Pearson correlation coefficient (df = 375) = 0.09, *p* = 0.062	0.92 (0.79–1.07) ^c^	
Gender (382) ^a^					
Male *	201	52.6	6.4 ± 2.8	94	47.7
Female	181	47.4	6.2 ± 2.7	103	52.3
			*t*-test (369) = 0.68, *p* = 0.492	1.49 (0.99–2.24) ^c^	
Firstborn (399) ^a^					
No	236	59.2	6.8 ± 2.7	69	34.9
Yes *	163	40.8	6 ± 2.8	129	65.1
			*t*-test (382) = −2.81, *p* = 0.005	0.65 (0.43–0.98) ^c^	
Chronic medical condition (401) ^a^					
No *	394	98.3	6.3 ± 2.8	195	97.5
Yes	7	1.7	7.5 ± 2.7	5	2.5
			*t*-test (384) = −1.17, *p* = 0.241	2.39 (0.46–12.51) ^c^	
Born preterm (400) ^a^					
No/Do not know *	349	87.2	6.2 ± 2.8	173	86.1
Yes	51	12.8	7.2 ± 2.1	28	13.9
			*t*-test (383) = −2.43, *p* = 0.015	1.28 (0.69–2.34) ^c^	
Ever been diagnosed with bronchiolitis (402) ^a^					
No *	353	87.8	6.1 ± 2.7	170	84.6
Yes	49	12.2	8 ± 2.5	31	15.4
			*t*-test (385) = −4.41, *p* < 0.001	1.97 (1.04–3.74) ^c^	
Ever been hospitalized for respiratory infection (401) ^a^					
No *	369	92	6.2 ± 2.8	180	89.5
Yes	32	8	7.9 ± 2.3	21	10.5
			*t*-test (384) = −3.35, *p* = 0.001	2.32 (1.03–5.21) ^c^	

^a^ Number of each item may not add up to the total number of the study population due to missing values. The number of respondents to each item is reported in parentheses, ^b^ mean ± standard deviation, ^c^ crude odds ratio (95% confidence interval), * reference category.

**Table 2 vaccines-12-00418-t002:** Results of the multivariate linear and logistic regression models.

**Variable**	**Coeff**	**SE**	**t**	** *p* **
Model 1. Parents’ concern that their child could get infected by RSV F (8, 315) = 17.11; R^2^ = 30.3%; adjusted R^2^ = 28.5%; *p* < 0.0001				
Having heard of RSV infection	2.05	0.31	6.48	<0.001
Child diagnosed with bronchiolitis	1.08	0.42	2.58	0.011
Need for additional information	0.71	0.29	2.41	0.017
Having acquired information about RSV from healthcare workers	0.72	0.36	2	0.047
Having more than one child	0.51	0.27	1.88	0.061
Both parents employed	0.31	0.29	1.06	0.292
Having a chronic medical condition	0.38	0.39	0.99	0.324
Older child	0.09	0.09	0.99	0.325
**Variable**	**OR**	**95% CI**		** *p* **
Model 2. Willingness to vaccinate their child against RSV Log likelihood = −149.19; *χ*^2^ = 143.97 (9 df); AUC * = 0.863; *p* < 0.0001 (sample size 322)				
Having heard of RSV infection	4.16	2.15–8.07		<0.001
Need for additional information	6.83	3.66–12.72		<0.001
Males	0.34	0.16–0.72		0.004
Having acquired information about RSV from healthcare workers	2.69	1.17–6.19		0.021
Female child	1.95	1.09–3.49		0.023
Both parents employed	1.92	0.98–3.74		0.055
Having more than one child	0.58	0.32–1.03		0.066
Having been hospitalized because of a respiratory infection	0.36	0.11–1.17		0.092
Both parents with a university degree	1.63	0.82–3.21		0.159

* Area under ROC curve.

## Data Availability

The data presented in this study are available upon request from the corresponding author.

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
