# Peer review of "Respiratory Syncytial Vaccination: Parents’ Willingness to Vaccinate Their Children"

_vaccines, 2024, doi:10.3390/vaccines12040418_

Round 1

Reviewer 1 Report

Comments and Suggestions for Authors

It was a good observation that 18.2% of parents were concerned about RSV vaccination and that their kids may get infected due to vaccination. One thing I don't understand between this and the information they received from HCWs.How the authors make them aware or create awareness for vaccination and giving vaccines is safe didn't mention those things here. So it would be helpful if authors would mention strategies to educate HCWs and parents about the role of vaccine and how it work. 

Author Response

Thank you for having reviewed the manuscript: “Respiratory Syncytial Vaccination: Parent's Willingness To Vaccinate Their Children”. We have fully addressed your concerns; below and in the new document we have indicated all the revisions made.

As suggested, we have included, in the Discussion section, more details about the strategies to educate HCWs and parents.

My colleagues and I are most grateful for the extremely positive tone of the reviewer’s comments.

Yours sincerely

Reviewer 2 Report

Comments and Suggestions for Authors

The purpose of this article was to evaluate the willingness of a group of parents to vaccinate their children against respiratory syncytial virus.
Therefore 404 people, mostly women, filled out a questionnaire divided into three sections.
The first aimed to gather socio-demographic information and health-related details of the relative;
the second consisted of 3 questions assessing participants' awareness of RSV infection;
the third comprised 3 questions regarding attitudes toward RSV infection and its vaccine.
Approximately 51% of parents expressed a willingness to vaccinate their children,
and among these the majority were well informed about the potential severity of the infection or wished to have more information.
The paper iw very interesting and well-described. Also the dscussion and conclusions are complete.
Only little questions:
Line 110: what is the Likert scale? Perhaps the geographical area in which the questionnaires were distributed is too limited

and could lead to a bias in the interpretation of the results.

Why were office workers more in favor of vaccination than non-employees?

Author Response

Thank you for having reviewed the manuscript: “Respiratory Syncytial Vaccination: Parent's Willingness To Vaccinate Their Children”. We have fully addressed your concerns; below and in the new document we have indicated all the revisions made.

1. The Likert scale, developed in 1932 by Rensis Likert to measure attitudes, is a ordinal scale usually and used by respondents to rate the degree to which they agree or disagree with a statement. (Likert R. A technique for the measurement of attitudes. Arch Psychology. 1932;22(140):55)

2. As suggested, in limitations paragraph we have added this sentence: “Moreover, the study  was carried out only in two regions of Southern Italy and this affect the generalizability of the findings.”

3. During the review, for a misunderstanding, we had to modify the statistical analysis and this variable is no longer associated with the outcome of interest.

My colleagues and I are most grateful for the extremely positive tone of the reviewer’s comments.

Yours sincerely

Reviewer 3 Report

Comments and Suggestions for Authors

I have some comments regarding the statistical analyses and the presentation of the results:

1. Outcomes: In chapter 2 (Materials and Methods), neither a primary nor a secondary outcome is specified. This information is rather important. The sample size calculation should be based on the primary outcome (i.e. willingness to vaccinate).

2. Quesionnaire: A self-administerd questionnaire was structured and tested in a pilot study. What conclusions have been drawn from this study? How has the validity of the questionnaire been checekd? - "No modifications ware made". What does this mean?

3. Results: Line 152: "... more than half had ONLY one child".  The word ONLY should be added (because each participant has at least one child).

4. Table 1: Instead of the ß coefficient, correlation coefficients according to Pearson should be given. Doing so, the p values will not change.

5. Table 2 (Model 1): How were the parameters in Table 2 selected? In chapter 2.4 you write "variables with a significance level of p <= 0.25 ... were incorporated". Looking at table 1, there are many variables with p <= 0.25; but only sex of them are presented in table 2. May be that you decieded to present only variables of the final multivariable model. However, then the question arises as to why variables are presented which are noch esignificant at all.

6. Table 2 (Model 3): See comments obove (table 2). 

7. "Both parents employed" may have a significant impact on the outcomes (tables 2 and 3). Howver, in table 1 only the factor "employed" is listed. That's doesn't go together. Something similar applies to "both parents with an university degreee".

8. "Need of additional education" in table 2 (model 3): It is very unlikely that p is ony a little bit more than 0.05 when OR is very close to 1. Please check this. Another question; Do you mean "no additional need of information" (model 3) or "Need of additional information" (table 1). This should be formulated uniformly.

9. Model 3: Furthermore: It is unlikely that factors in the multiple analysis have a much lower p value compared to univraible analysis (i.e. males, firstborn child, female child). Can you explain this?

10. In table 2, models 1 and 3 are rpesented. Waht about model 2?

11. For model 1, R2 is given as a meausre of goodness. Accordingly, yo should presend AUC for model 3.

12. Page 8, line 244: You list four independent determinants which are allegedly significant for both models. Howver, in model 1 "need of additional information" is not signifcant (it doesn't appear in table 2) whereas "Having a child diagnosed with bronchitiolitis" is not significant in model 3.

13. Having a daughter may be a predictor of parents' willingness to vaccinate their child. That seems to be a bit strange. Do you have an explanation for this?

Comments on the Quality of English Language

The quality of English Language is fine.

Author Response

Thank you for having reviewed the manuscript: “Respiratory Syncytial Vaccination: Parent's Willingness To Vaccinate Their Children”. We have fully addressed your concerns; below and in the new document we have indicated all the revisions made.

1: As suggested, we have specified, in statistical analysis section, that primary outcome was willingness to vaccinate their child against RSV and secondary outcome  was parents’ concern that their child could get infected by RSV. Moreover, as already specified in Setting and Study Population section, the sample size has been calculated on the primary outcome: “willingness to vaccinate their child against RSV”.

2: As suggested, a better explanation of the pilot study has been reported in the Questionnaire paragraph.

3: As suggested, we have added the word “only”.

4: As suggested, we have added the correlation coefficients according to Pearson.

5: In response to the point regarding how variables were presented in the models, the p value between 0.2 and 0.4 has been used for the stepwise regression analysis. Indeed, the criteria for our statistical model building for the regression models involves seeking the most parsimonious model that still explains the data, so that the resultant model is more likely to be numerically stable and more easily generalized. For the selection of variables, we have used the well-known strategy suggested by David W. Hosmer Jr., Stanley Lemeshow, Rodney X. Sturdivant (Model-building strategies and methods for logistic regression. In: Applied Logistic Regression. John Wiley & Sons, New York; 2013;89-151). Indeed, the second step was to fit the multivariate models and, therefore, we constructed stepwise multivariate linear and logistic regression models and the significance level for variables entry each model was set at 0.2 and for removal at 0.4. Moreover, this approach was similar to the methodology that has been used in many other studies conducted by some of us (for example, Vaccines (Basel) 2022;10:777; Vaccines (Basel) 2022;10:396; Vaccines (Basel) 2022;10:146; Expert Review of Vaccines 2022;21:541-547; Expert Review of Vaccines 2021;20:881-889).

6: The previous answer also explains how variables were presented in Model 2.

7: As suggested, we have standardized the variable name.

8: As suggested, we have verified and modified the statistical analysis. Moreover, we have updated the results in the results section and formulated uniformly the variable name.

9: As suggested, we have verified and modified the statistical analysis and updated the results.

10: As suggested, we have changed “Model 3” in “Model 2”.

11: As suggested, we have added the AUC for model 2 in table 2 and reported the graph below.

12: As suggested, in the Discussion section we have rephrased and clarified this point.

13: In response to this point, we have specified in the Discussion section that “gender differences in vaccination uptake in the adult population has been reported, and attention to gender-related issues in immunization programs, with the goal of reducing gender-related barriers to access to vaccination specifically addressed to female children, is considered a priority in the international literature, specifically in low income countries. The results of this study run counter to the literature and are worth of further investigation.”

My colleagues and I are most grateful for the extremely positive tone of the reviewer’s comments.

Yours sincerely

Reviewer 4 Report

Comments and Suggestions for Authors

The paper is interesting and well-written.  I have a few suggestions for the authors:

1. Introduction: 

It is unclear whether the authors intend for the vaccine to be included as part of the vaccination program or if parents will be required to vaccinate at their own expense.

2. Methods:

* How were the questionnaires distributed to parents?

*Questionnaire - Provide details on the construction of the questionnaire. Who wrote it, and how was it validated? Please attach the questionnaire as an appendix.

* On what basis did the authors determine the significance level as p<0.25?

Discussion:

There is an additional limitation - the study was conducted in a specific region in Italy and seems non-representative. This hinders the generalizability of the findings to the overall population.

Conclusions:

Suggestions for future research in the field should be added.

Author Response

Thank you for having reviewed the manuscript: “Respiratory Syncytial Vaccination: Parent's Willingness To Vaccinate Their Children”. We have fully addressed your concerns; below and in the new document we have indicated all the revisions made.

1: As suggested request, in the introduction section we have clarified this point.

2: As suggested, we added more details on where the questionnaire was distributed

3: As suggested, in the questionnaire paragraph we have added more details about the construction of the questionnaire. The questionnaire has been attached as an appendix.

4: In response to the point regarding why we used a p-value ≤ 0.25 as threshold in the univariate analysis, the multiple logistic regression models were built using the strategy suggested by Hosmer et al. [Hosmer JrDW, Lemeshow S, Sturdivant RX. Applied Logistic Regression.NewYork, NY: JohnWiley& Sons (2013)]. According to this well-known model-building strategy, through the use of univariable analyses it is possible to identify, as candidates for a multivariable model, any variable whose univariable test has a p-value less than 0.25.

5: As suggested, this point has been added in the Limitations paragraph.

6: As requested, suggestions for future research in the field has been added in the Conclusions section.

My colleagues and I are most grateful for the extremely positive tone of the reviewer’s comments.

Yours sincerely

Round 2

Reviewer 3 Report

Comments and Suggestions for Authors

My comments and suggestions for improvement were adequately implemented. In this revised revision, the manuscript can be published.